# A Review of the Asymmetric Numeral System and Its Applications to Digital Images

**DOI:** 10.3390/e24030375

**Published:** 2022-03-07

**Authors:** Ping Ang Hsieh, Ja-Ling Wu

**Affiliations:** 1Graduate Institute of Networking and Multimedia, Taipei 10617, Taiwan; r08944039@csie.ntu.edu.tw; 2Department of Computer Science and Information Engineering, National Taiwan University, Taipei 10617, Taiwan

**Keywords:** entropy encoding, Huffman coding, arithmetic coding, Asymmetric Numeral System, joint image compression and encryption

## Abstract

The Asymmetric Numeral System (ANS) is a new entropy compression method that the industry has highly valued in recent years. ANS is valued by the industry precisely because it captures the benefits of both Huffman Coding and Arithmetic Coding. Surprisingly, compared with Huffman and Arithmetic coding, systematic descriptions of ANS are relatively rare. In 2017, JPEG proposed a new image compression standard—JPEG XL, which uses ANS as its entropy compression method. This fact implies that the ANS technique is mature and will play a kernel role in compressing digital images. However, because the realization of ANS involves combination optimization and the process is not unique, only a few members in the compression academia community and the domestic industry have noticed the progress of this powerful entropy compression approach. Therefore, we think a thorough overview of ANS is beneficial, and this idea brings our contributions to the first part of this work. In addition to providing compact representations, ANS has the following prominent feature: just like its Arithmetic Coding counterpart, ANS has Chaos characteristics. The chaotic behavior of ANS is reflected in two aspects. The first one is that the corresponding compressed output will change a lot if there is a tiny change in the original input; moreover, the reverse is also applied. The second is that ANS compressing an image will produce two intertwined outcomes: a positive integer (aka. state) and a bitstream segment. Correct ANS decompression is possible only when both can be precisely obtained. Combining these two characteristics helps process digital images, e.g., art collection images and medical images, to achieve compression and encryption simultaneously. In the second part of this work, we explore the characteristics of ANS in depth and develop its applications specific to joint compression and encryption of digital images.

## 1. Introduction

In our review paper, we present the operational details and possible applications of the newly developed lossless compression algorithm—Asymmetric Numeral System (ANS). ANS is one of the most recently proposed entropy coding methods. Fast execution speed and close to the theoretical limit compression performance are the prominent features of ANS; therefore, it has been primarily adopted by industrials. Jarek Duda first proposed ANS in 2007 [1,2,3], and it was adopted and implemented by Facebook in 2015, namely, Zstandard [4], which is open-sourced and used in various fields such as Linux Kernel/Hadoop/Mysql/FreeBSD. Apple also released its ANS implementation—LZFSE [5]—in 2015 and used it at the bottom layer of iOS and macOS. Google launched its lossless compression standard—pik [6]—in 2019, in which the entropy coding part also uses ANS. Microsoft also applied for ANS-related patents [7] in 2019. In addition to the industry giants mentioned above, the JPEG standard committee began drafting the new compression standard JPEG XL [8] in 2017. ANS also plays a significant role in its entropy coding. We can see that in the past five years, ANS has been widely accepted and adopted by the IT giants, but in the compression academia community and nonexpert IT industry, the awareness and the adoption of ANS for Multimedia compression is still in its infancy.

Its lossless compression feature makes ANS especially suitable for distortion-less compression-related applications, such as medical and digital art collection images. The prospective property of ANS comes from its chaotic characteristics: if the original input is changed a little bit, its compressed output will change relatively significantly. Similarly, if we slightly change the compressed representation, the reconstructed version will also present a rather significant change after decompression. This kind of significant range’s difference between input and output of a function is one of the preferred features and is called the avalanche effect in cryptography [9]. Recall that, in ANS, encoding an input symbol will produce two outputs: a positive integer state and a segment of a bit sequence (we call this the segmentation feature of ANS). As mentioned above, if the input changes a little, the corresponding integer state and the bitstream segment of the output will change significantly. Conversely, if we tiny modify the integer state or the bitstream segment of the ANS production, the reconstruction will also significantly change after decompression. The avalanche feature mentioned above is suitable for providing a compact representation of digital art collection images. A digital art image is now represented by a positive integer state and a bitstream sequence. Art collectors can store the state separately and open it to the public as a piece of evidence for claiming the ownership of this artwork while keeping the bitstream sequence in private as the verifier if a dispute occurs. Because of its avalanche characteristics, we think there will be an excellent opportunity to combine ANS with the recently popular NFT (Non-fungible token) [10] to make the intellectual property rights (IPRs) of an artwork much more secured.

With ANS’s segmentation feature, we can assign different degrees of protection to various portions of an artwork according to their art values. For example, the portrait in the middle of the Mona Lisa image certainly has higher art value than its corners or other flat counterparts. An artwork publisher who intends to sell his digital artworks to more than one artwork collector can divide his art collection into different pieces and price them according to the corresponding values. Now, combining all specific features of ANS, the publisher can generate the state and the bit sequence for each partition. He can now disclose the state information to the potential customers as a marketing representative of this partition in NFT applications. Moreover, the publisher can send the bit sequence of the same segmented area to the actual buyer as a voucher for certifying the ownership. We will justify the above postulation through a concrete experiment, with the aid of table-ANS [11], at the end of this writeup. 

The contributions of this work include

We present an in-depth and systematic discussion about various ANS-related technologies for providing a clear picture of this new lossless compression tool;We address several selected applications of ANS in response to the survey nature of this work;We explore the chaotic property of ANS and apply it to compress and encrypt digital images jointly, which is the desired mechanism for most digital image generators;We present a detailed performance comparison of various lossless compression algorithms in terms of compression ratio and execution speed.

In addition, as application examples, we will explore the feasibility of using ANS to protect IPRs of art collection images and check the integrity of medical images.

## 2. Background Knowledge

### 2.1. Basic Concepts of Asymmetric Numeral Systems

An ANS coder will encode an input to a non-negative integer number and call it the state. Mathematically, we can illustrate the ANS encoding process as follows.

ANS-encoding: (input, current state) → (next state)ANS-decoding: (current state) → (previous state, output).

That is, using the language of the Finite State machine, ANS encoding can be realized as a transition from a given current state to its next state. At the same time, the ANS decoding process plays the reverse role of the encoding process (cf. Figure 1). 

Therefore, as shown in Figure 1, we can regard ANS encoding and decoding processes as state transitions on a Finite State Machine. Each node denotes a legal state (with an integer state value). Furthermore, according to the symbol ‘a’, each edge transits from one node to another.

### 2.2. Huffman Coding, Arithmetic Coding, and the Asymmetric Numeral Systems

Huffman Coding [12] and Arithmetic Coding [13] are the most well-known and adopted algorithms among the entropy compression methods. As described, ANS is the newest entropy coder that the industry has highly valued in recent years. ANS is valued by the industry precisely because it captures the benefits of both Huffman Coding and Arithmetic Coding [2]. Huffman Coding is known for its fast encoding and decoding but has limitations in compression performance (at least one bit is required to represent a symbol). On the contrary, Arithmetic Coding is characterized by a high compression ratio (the degree of compression can be close to the theoretical optimal value) but has limitations in encoding and decoding speed. 

Generally speaking, the slow execution speed disadvantage of Arithmetic Coding comes from its involvement in floating-point numbers calculations, which complicates the practical realization and slows down the entire compression and decompression process. The shortage mentioned above of arithmetic codes can be understood as follows. Theoretically, the amount of self-information contained in a symbol s with probability $p_{s}$ is $\log_{2}\left(\frac{1}{p_{s}}\right)$ bits. Similarly, in conventional arithmetic coding, the amount of self-information for two continuous coding stages *x* and $x^{\prime }$ will be $-\log_{2}p_{x}$ and $-\log_{2}p_{x^{\prime }}$ bits, respectively. After transition from stage *x* to stage $x^{\prime }$, by encoding the new symbol s; ideally, we have $p_{x^{\prime }}=p_{x}p_{s}$. Therefore, in arithmetic codes, the probability range after encoding the incoming symbol shrinks from the previous range by multiplying the probability of the symbol s, which is less than 1. This explains why floating-point numbers are used in arithmetic coding’s implementation. To overcome this shortage, as one of the anonymized reviewers mentioned, modern Arithmetic Coding implementations use renormalization, which helps avoid floating-point operations. The first such fully integer multi-symbol implementation of Arithmetic Coding was proposed in 1987 in [14]. Nevertheless, the implementation in [14] needs multiplications and divisions; therefore, several look-up table-based adaptive binary arithmetic coding implementations were proposed, which many video and image compression standards have adopted. Moreover, there is more advanced research work related to adaptive range coding (Arithmetic Coding with fast renormalization), for example [15], and multiplication and division free multi-symbol Arithmetic Coding [16]. 

Different from the prescribed speeding up approaches for Arithmetic Coding, to speed up the processing speed, in ANS, a positive integer state value is the desired target. To achieve this goal, instead of shrinking the new state variable’s range, Jarek Duda [1] suggested dividing the original state variable’s range by the symbol’s probability to expand it into integer values, that is $x^{\prime }\approx \frac{x}{p_{s}}$. Therefore, if s ∈ {0,1}, each state transition doubles the original state range, while if s ∈ {0, 1, 2, …, 9}, each state transition will ten times enlarge the new state’s range. This kind of assignments, in some sense, make the behavior of ANS similar to that of the conventional weighted number systems, such as Binary and Decimal number systems.

### 2.3. Types of the Asymmetric Numeral Systems

According to the distributions of the source symbols and methods of realization, there are three variants of ANS [1,2,3,17,18,19,20,21,22,23,24,25,26,27,28,29,30,31,32,33,34] (follow the chronological sequence of publication dates):(i)Uniform Asymmetric Binary System (uABS)(ii)Range Asymmetric Numeral System (rANS)(iii)Table Asymmetric Numeral System (tANS)

We will present the definitions and operating processes of various ANSs in the rest of this section, in “learning by examples” and “step-by-step” ways. We will explain different versions of ANS encoding and decoding procedures in detail through concrete examples. Before going into details, a summary about the characteristics of different types of ANS is given as follows. The inputs processed by uABS are only 0 and 1. The information processed by rANS is not only 0 and 1 but with a variety of possibilities. tANS tabularizes the ANS’s encoding and decoding processes.

## 3. Variations in Asymmetric Numeral Systems

### 3.1. The Uniform Asymmetric Binary System (uABS)

uABS is the most basic type, and the input processed by it is only two possible cases: 0 or 1. Expressed by a mathematical formula, the input set A looks like: $A=\left\{0,\;1\right\}≜\left\{s_{0},\;s_{1}\right\}$, with probability distributions: $p\left(s_{0}\right)=p_{0}=p\;,p\left(s_{1}\right)=p_{1}=1-p,\;and\;p_{0}+p_{1}=1$. In uABS, the input is a series of finite number bitstreams consisting of 0 or 1, such as 010011. The output will be a natural number (i.e., a non-negative integer). For simplicity, we use x to denote the state variable of a node. Therefore, in the encoding process, as mentioned above, state transitions are performed as Enc (input bit, current state) → (next state); or symbolically reduces to C (s, x) = x′. We also use state transitions to realize the decoding process: D (x′) = (x, s).

(a)uABS Constructions for Uniformly Distributed Binary Sources

As described in Section 2.2, the function of an uABS (or an ANS in general) encoder can be represented: (1)$$C\left(s,\;x\right)=x^{\prime }\approx \frac{x}{p_{s}}$$
This arrangement shows that the smaller the probability of the symbol encoded, the larger the new state number (or state-variable range) after state transition. This implies that if the probability of the current encoding symbol is smaller, then we need more bits to represent its corresponding uABS output. 

To give readers a clear picture of the process of ANS encoding, let us examine the following simple example first.

Example 1: Assume *s* = {0,1} and *p*_0_ = *p*_1_ = 0.5. According to Equation (1), the best encoding function for 0 or 1 would be $C\left(0,\;x\right)=C\left(1,\;x\right)=\frac{x}{p_{0}}=2x$. In fact, taking the polarity of input symbols into account, the encoding function becomes
(2)$$C\left(s,\;x\right)=x^{\prime }=2x+s$$
and the decoding function is
(3)$$D\left(x^{\prime }\right)=\left(x,\;s\right)=\left(\lfloor \frac{x^{\prime }}{2}\rfloor ,\;\;x\;mod\;2\right)$$
Now, for the input sequence $b_{1}b_{2}b_{3}b_{4}b_{5}=01111$, the initial state is $x_{0}=1,$ and the Encoding process is conducted in order as follows:$$C\left(b_{1},\;x_{0}\right)=x_{1\;}=2x_{0}+b_{1}=2*1+0=2\;C\left(b_{2},\;x_{1}\right)=x_{2\;}=2x_{1}+b_{2}=2*2+1=5\;C\left(b_{3},\;x_{2}\right)=x_{3\;}=2x_{2}+b_{3}=2*5+1=11\;C\left(b_{4},\;x_{3}\right)=x_{4\;}=2x_{3}+b_{4}=2*11+1=23\;C\left(b_{5},\;x_{4}\right)=x_{5\;}=2x_{4}+b_{5}=2*23+1=47.$$
That is, for the input $b_{1}b_{2}b_{3}b_{4}b_{5}=01111$, the corresponding uABS output is the positive integer 47, which is also the value (or range) of the state variable $x_{5\;}$.

Similarly, the relevant Decoding process is conducted in order as follows:$$D\left(x_{5}\right)=\left(x_{4},\;b_{5}\right)=\left(\frac{x_{5}}{2},\;x_{5}\;mod\;2\right)=\left(\frac{47}{2},\;47\;mod\;2\right)=\left(23,\;1\right)\;D\left(x_{4}\right)=\left(x_{3},\;b_{4}\right)=\left(\frac{x_{4\;\;}}{2},x_{4}\;mod\;2\right)=\left(\frac{23}{2},\;23\;mod\;2\right)=\left(11,\;1\right)\;D\left(x_{3}\right)=\left(x_{2},\;b_{3}\right)=\left(\frac{x_{3\;}}{2},x_{3}\;mod\;2\right)=\left(\frac{11}{2},\;11\;mod\;2\right)=\left(5,\;1\right)\;D\left(x_{2}\right)=\left(x_{1},\;b_{2}\right)=\left(\frac{x_{2\;\;}}{2},x_{2}\;mod\;2\right)=\left(\frac{5}{2},\;5\;mod\;2\right)=\left(2,\;1\right)\;D\left(x_{1}\right)=\left(x_{0},\;b_{1}\right)=\left(\frac{x_{1\;\;}}{2},x_{1}\;mod\;2\right)=\left(\frac{2}{2},\;2\;mod\;2\right)=\left(1,\;0\right).$$
Clearly, for input 47, the uABS output would be $b_{1}b_{2}b_{3}b_{4}b_{5}=01111$. It is the same as the original input.

For speeding up the whole coding process, table lookup techniques are often used in entropy coding areas. This convention also applies to ANS. In the following, we will use a so-called coding table to illustrate the encoding and decoding processes of uABS with uniformly distributed inputs.

Example 2. Assume *s* = {0,1} and *p*_0_ = *p*_1_ = 0.5. Let us consider the following coding table, where the table occupancy of 0 and 1 is the same since they have the same probability distribution.

First, let us take the red-3 in the bottom row of the Table 1 as an example to explain from the perspective of encoding. Assume the red 3 is the current state, then from the index of the row it belongs to, we say that the symbol to be encoded is *s* = 1. In contrast, the corresponding column denotes the encoded state *x′* = 7. According to our previous discussions, mathematically, we have the following uABS expression: $C\left(s,\;x\right)=x^{\prime }\Rightarrow C\left(1,\;3\right)=7$. Second, let us continue to use the red 3 as an example to explain from the perspective of decoding. Now, the red 3 represents the decoded state x, and its corresponding row index denotes the decoded symbol *s* = 1. The corresponding column shows that the to-be-decoded state is *x′* = 7. Mathematically, its uABS expression becomes $D\left(x^{\prime }\right)=\left(s,\;x\right)\Rightarrow D\left(7\right)=\left(1,\;3\right).$ In this way, as long as we know the coding table, we can completely describe the encoding and decoding processes very efficiently. However, the question is: ‘how is the coding table constructed?’ We will answer this question later.

(b)ABS Constructions for Non-Uniformly Distributed Binary Sources and the Symbol Spread Function

From Example 2, we can find an interesting phenomenon: when the symbol to be encoded is 0, the generated next state *x′* is an even number, and when the symbol is 1, the next state *x′* is an odd number. The reason comes from the encoding function C (*x, s*) = *2x* + *s*. Therefore, depending on the polarity of *s*, we can divide the coding states into two categories: even-numbered and odd-numbered types.

This observation reveals that there is an allotting mechanism between a given symbol and its possible mapping states. This mapping mechanism plays an essential role in building efficient and effective realization of ANS, which is called the symbol spread function (SSF) [1]. Simply put, SSF addresses the mapping relation from states to symbols. Here, we use the notation s¯ to represent the symbol spread function. Mathematically, we have the following expression: s¯: ℕ→A⇒ s¯(x)=s, where ℕ denotes the set of natural numbers, and A is the set of involved source symbols. With this notion, the SSP used in the above two examples can be written as s¯(x)=x mod 2. 

In the physical sense, SSF divides a given state into several subsets and allocates a different symbol to each distinct subgroup. Since ANS is mainly applied to compress data, therefore, the effectiveness of an SSF is judged by its compression performance and execution speed. Unfortunately, finding the best SSF for an ANS construction involves solving complicated combinatorial problems; therefore, sub-optimal heuristic approaches are adopted in most practical use cases.

From our previous discussions, the selection of SSF is closely related to the symbol probability $p_{s}$. Moreover, the encoding function $C\left(s,\;x\right)=x^{\prime }\approx \frac{x}{p_{s}}$ tells us that the encoded state *x′* corresponding to symbol s would appear at an integer multiple of intervals with spacing $\frac{1}{p_{s}}$, in a tabularized realization non-uniformly distributed ABS, since the input state *x* here could be any non-negative integer. 

In summary, if the probability of symbol *s* is more significant (its occupancy is higher and the symbol appears more times in the coding table). In addition, the interval spacing between neighboring states $x^{\prime }$ will be smaller, which means more states will be allocated to the same symbol *s*. The physical meaning is that the ANS hopes to give more states for the symbol that appears more often. To give readers a clear picture of the process of non-uniform ABS encoding, we give an illustrative simple in Appendix A.

We take non-uniform input 01111 as another illustrative example to further discuss this scenario. In this case, the probability of 0 is $\frac{1}{5}$ and the probability of 1 is $\frac{4}{5}$ As we can see in the following Table 2, the appearance of 0 is one out of five, and the appearance of 1 is four out of five.

### 3.2. The Range Asymmetric Numeral System (rANS)

(a)The Basic rANS Construction

In an rANS, the set of input symbols to be encoded is $A=\left\{s_{0},\;s_{1}\ldots s_{n}\right\}$, and the number of occurrences of each symbol $s_{i}$ is $L_{i}$, the total number of occurrences of all symbols is L, $L=\sum L_{i}$. Assume the probability of symbol $s_{i}$ is $p_{i}$, $p_{i}=\frac{L_{i}}{L},\;and\;\sum p_{s}=1$. In the following discussions, we call $L_{i}$ a sub-cycle, L a cycle, and ‘cycle’ also stands for ‘the range’ of an rANS. The major difference between rANS and uABS is whether the number of symbols involved in the process is more than two or not. To explain the concept of rANS, again, we start with a simple example.

Example 3. Suppose *A* = {*a, b, c*}, with probability distributions: $p_{a}=\frac{5}{8},\;p_{b}=\frac{2}{8},\;and\;p_{c}=\frac{1}{8}$. Notice that this assumption of symbol distributions is the same as in Figure 3 of [35]; therefore, the same repeating patterns are obtained, as shown in Figure 2. From the above discussions, the ideal SSF for this example should assign symbol *s* to state *x′* consistently according to $p_{s}$. So, symbol a should occupy $\frac{5}{8}$ of all states, symbol b should occupy $\frac{2}{8}$ of all states, and symbol c should occupy $\frac{1}{8}$ of all states. According to this concept, we have the following coding table.

Of course, this deduction still applies to cases with other different probability distributions, as shown in Figure 2.

Following the same inferencing, the proper SSF for Example 3 would be: $$\bar{s}\left(x\right)=\{\begin{array}{c} a,\;\;\;mod\left(x,8\right)=\left\{0,\;1,\;2,\;3,\;4\right\}\\ b,\;\;mod\left(x,\;8\right)=\left\{5,\;6\right\}\\ c,\;mod\left(x,8\right)=\left\{7\right\} \end{array}$$
Or, we can express $\bar{s}\left(x\right)$ as a repeated pattern ‘aaaaabbc’ with period 8. Similarly, it is easy to find that the encoding functions $C\left(a,\;x\right)=\frac{x}{p_{a}}=\frac{8}{5}x$, $C\left(b,\;x\right)=\frac{x}{p_{b}}=\frac{8}{2}x$, and $C\left(c,\;x\right)=\frac{x}{p_{c}}=8x$ do not work well. In the next paragraph, we will pay attention to derive the actual encoding functions for Example 3. 

First, let us define the Cumulated Distance Function, $CDF\left[s\right]=\sum L_{S^{\prime }}$; its physical meaning is to find the sum of the sub-cycle lengths of the symbol *s’* before the to-be-encoded symbol *s*, in a cycle. For example, if the to-be-encoded symbol is b, then *s’ = a*, and CDF[*b*] = sum of the sub-cycle lengths for symbol *a* = 5. Second, since there is more than one sub-cycle in the coding table for the given symbol *s*, according to the current state *x*, we can find which sub-cycle the to-be-encoded symbol *s* belong to simply by calculating $\lfloor \frac{x}{L_{s}}\rfloor +1,$ where ⌊y⌋ denotes the largest integer less than *y*. For example, if we are computing *C(b,3)*, then $\lfloor \frac{3}{L_{b}}\rfloor +1$ = 2 tells us that we are now encoding that symbol *b* in its second sub-cycle. Moreover, $\left\lfloor \frac{x}{L_{s}}\right\rfloor$ ∗ L = $\left\lfloor \frac{3}{L_{b}}\right\rfloor$ * 8 = 8 means we should add a bias 8 to calculate the address of the next state *x′*. Finally, we should now find the exact position of the current state *x* in the sub-cycle it belongs to, and computing x mod $L_{S}$ can quickly achieve this goal. Therefore, combining all the relevant calculations we have
$$\begin{array}{ll} C\left(b,\;3\right) & =\lfloor \frac{3}{L_{b}}\rfloor \times L+CDF\left[b\right]+3\;mod\;L_{b}\\ & =\lfloor \frac{3}{2}\rfloor \times 8+CDF\left[b\right]+3\;mod\;2\\ & \;=8+5+1=14 \end{array}$$
From the above discussions, we can conclude that the proper ANS encoding and decoding functions for a non-binary source with different symbol probability distributions should respectively be:(4)$$C\left(s,\;x\right)=\lfloor \frac{x}{L_{s}}\rfloor \times L+CDF\left[s\right]+x\;mod\;L_{s}$$
(5)$$D\left(x^{\prime }\right)=\left(x,\;s\right)=\{\begin{array}{c} x=\lfloor \frac{x^{\prime }}{L}\rfloor \times L_{s}-CDF\left[s\right]+x^{\prime }\;mod\;L\\ s=x^{\prime }\;mod\;L \end{array}$$
According to Equation (4), the rANS coding table for Example 3 should be shown in Table 3.

Notice that the main difference between Table 3 and Table 4 lies in the number of encoded states x’. Since there are three distinct symbols and the minimum symbol probability is one-eighth; therefore, as shown in Table 3, there are 24 states in total. Moreover, we can easily check the correctness of Equation (5) by computing: D (14) = $\lfloor \frac{14}{8}\rfloor \times 2-CDF\left[b\right]+14\;mod\;8$ = 2 − 5 + 6 = 3 and s is the (14 mod 8 =) 6-th symbol in the coding table, which is b. Since all the above derivation is based on the symbol’s range occupation in the coding table, we think this is why this approach is called the range ANS in the literature. 

To accelerate the decoding speed, besides the above addressed basic coding table construction, the period of the repeat pattern (or the sum of the sub-cycle lengths), *L*, is usually selected as an integer power of 2, that is $L=2^{n}$. With this setting, in the decoding, we can use bit-shifting instead of division to realize $\lfloor \frac{x^{\prime }}{L}\rfloor \times L_{s}$ and use masking instead of modular operation to implement $x^{\prime }\;mod\;L$. In this way, a decoding process needs one multiplication operation only.

(b)Streaming ANS Coding and the Renormalization Process

The two ANSs discussed earlier, uABS and rANS, face a common serious problem: the state value will become larger and larger in the encoding process if a streaming (or continuous) data source is encountered. This unbounded growth of state range is unacceptable in practice because, in any computer architecture, the realizable integer is always limited. For example, in a 64-bit computer, the largest type of integers is the Unsigned Long Int, and its range is [0, $2^{64}-1$]. If we want to encode an ultra-long sequence, there will be an overflow even the largest integer type is adopted. In contrast, along with the decoding process, the state value will decrease and eventually be smaller than 0 and jump to a negative integer number. In addition, the negative integers have their limits on a computer.

To keep the state values within the computer representable integer ranges during the encoding and the decoding processes, we should derive a dynamic mechanism for adjusting the state ranges during the coding processes. When the state value is less than the allowable range, the mechanism will increase the state range accordingly and vice versa. We call the state range adjusting mechanism the renormalization process in ANS. 

Before defining the renormalization process, we noticed that although both the ANS encoding and decoding involve state transitions, they cannot be described correctly by a Finite State Machine because the involved states have unbounded ranges if a streaming source is considered. In other words, the numbers of possible state ranges become finite only after applying the renormalization process. This bounded involved state range makes the corresponding ANS realizable by using a limit-sized computational facility. Since both in encoding and decoding, the involved states may exceed the allowable ranges of the computing device, we will discuss the renormalization mechanisms for the encoding and the decoding processes, respectively.

In ANS encoding, when the state value goes out of the designated range, the renormalization process shifts the out-of-range state value one bit to the right, that is, divide the state value by 2, then removing the least significant bit (LSB) from the state value and stuffing it into the newly defined ‘ANS-bitstream variable.’ For example, suppose the designated ANS state range is [18,32]. Now, if the next encoding state goes to 70, which exceeds the maximum allowable range of 29, since the binary representation of 70 is 1000110_2_, and after shifting one bit to the right, we have 100011_2_ = 35, which is still larger than 29. So, we move one bit of 35 to the right again and obtain 10001_2_ = 17 within the target range. Of course, the two right-shifted bits 10 are now stored in the pre-described ANS-bitstream variable, and the renormalization ends. With the aid of renormalization, we can continuously encode the incoming source symbols and guarantee the state range is within the predefined bound. For ease of understanding the prescribed renormalization mechanism, Appendix B presents the pseudo-codes and illustration examples for both the ANS Stream Encoding and the ANS Decoding.

Observing the extreme example presented in the latter part of Appendix B, we can conclude that: to guarantee the proper operation of ANS Stream Coding, the total number of states in the allowable state range must be larger than the number of involved source symbols. Thus, the compactness consideration gives us the best choice of UI_s_ = b × L_s_. Recall that in ANS, to speed up the processing speed, expanding the new state range by dividing the original state range by the symbol’s probability is used instead. This statement tells us that the lower bound of the allowable state range, ILs, is determined by the smallest probability of the source symbol, which may bring challenges in realization when the source vocabulary is enormous. Fortunately, [32] investigated how to extend ANS’s capability to serve the situation that the size of the input set is considerably large—thousands or millions of symbols. Under this condition, the table size for realizing tANS will be huge also. This new situation has not been addressed in the traditional ANS-related research. Most of the ANS-related studies dealt with unsigned byte (uint8_t) inputs, but [32] deals with unsigned integer (uint32_t) inputs and even higher precision cases. The most significant contribution of [32] comes from its discussion and investigation about finding a reasonable and realizable capability. Moreover, [32] proposes ways to achieve the maximal allowable capacity based on symbol folding and Partial Alphabet Re-Ordering. The core idea of symbol folding lies in a particular coding technique—Elias gamma coding, commonly used when coding integers whose upper bound cannot be determined beforehand. This characteristic fits well with the new condition (i.e., the size of the input set can be enormous). The core idea of Partial Alphabet Re-Ordering is to supplement the case that symbol folding cannot do well—the most frequent symbols have the high symbol number. [32] took the technique to enhance byte codes with restricted prefix properties proposed by S. Culpepper and A. Moffat [36] in 2005 to surpass this challenging case. As shown in [32], with the aids of two existing techniques, we can apply ANS to handle applications with an extensive involved alphabet set. 

For ease of discussion, let us focus on the case of a source with reasonable source symbols in the rest of this work. After understanding why we design the allowable state range in this way, we start to apply the renormalization process to rANS for making it feasible in practice in the following sub-section.

### 3.3. The Table Asymmetric Numeral System (tANS)

As the name suggests, tANS focuses on the subject of ANS’s realization using lookup tables. That achieves all encoding and decoding operations through table lookups, making encoding and decoding faster and easing for hardware implementations [17,18,31,33]. Since all processes are operated in a table, the size of the table must have a limit, and this is equivalent to set an upper bound on the number of states. This design thinking is the same as that of the stream rANS discussed earlier.

Similar to rANS, in tANS, I :={L, L+1, …, 2L−1}, where $L=2^{R}$, R is a positive integer. For each symbol *s*, its state range $I_{s}$ is $I_{s}\;:=\left\{L_{s},\;L_{s}+1,\;\ldots ,\;2L_{s}-1\right\}$, where $L_{s}$ is the number of occurrences of symbol *s*. Assume the actual probability of symbol s is $p_{s}$, $\frac{L_{s}}{L}=q_{s},$ and $q_{s}$ is designed as close to $p_{s}$ as possible. The same as with the conventional entropy coding, the higher the difference between $q_{s}$ and $p_{s}$, the worse the compression efficiency.

Recall from Example 3, the corresponding SSF is $\bar{s}\left(x\right)$ = aaaaabbc, which is an orderly arrangement. In tANS, $\bar{s}\left(x\right)$ can be arranged in much more ways; for example, $\bar{s}\left(x\right)$ = abaaabaac is another proper choice. Actually, for this particular example, the total number of possible $\bar{s}\left(x\right)$ will be $\frac{8!}{5!2!1!}=168$, and this is only an example with a fairly small number of source symbols. Generally speaking, when an English file is to-be-compressed, ASCII code is the most often used symbol representation. That is, a symbol has 256 possibilities. In this setting, all possible numbers of SSF $\bar{s}\left(x\right)$ are $\frac{256!}{i_{1}!i_{2}!\ldots i_{n}!},\;where\;i_{1}+i_{2}+\ldots +i_{n}=256$, and the number of possible choices is relatively large. Therefore, the SSFs of tANS provide more possibilities for encoding/decoding, which increases the degree of system chaos and provides more vital cryptographic characteristics. Moreover, the associated broader choice in SSF also offers more room for optimizing the compression performance. It follows that the proper design of an SSF plays the core role in tANS. 

(a)The Encoding and Decoding Functions of tANS:

Due to their similarity in behavior, the design of tANS follows the same principles of the rANS stream encoder. From the pseudo-codes of ANS stream encoding presented in Appendix B, we found a while loop in it. At first glance, it seems this while loop will run for a long time, but in fact, we can use *O(1)* time to calculate how many while loops we need to run in advance, as follows.

Assume the to-be-encoded symbol is *s,* and the current encoded state value *x* is higher than the upper bound of the designated allowable state range. According to the renormalization principle, we must shift the current state *x* to the right several times to constrain the resulting state value within the target state range.

Let $k_{s}\left(x\right)$ denote how many times the while loops we need to run in advance. For a given target state range $I_{s}:=\left\{L_{s},\;L_{s}+1,\;\ldots ,\;2L_{s}-1\right\}$ it is easy to derive $k_{s}\left(x\right)=\log_{2}\lfloor \frac{x}{L_{s}}\rfloor$. After knowing $k_{s}\left(x\right)$, we will modify the calculations of mod(x, 2) to $mod\left(x,\;2^{k_{s}\left(x\right)}\right)$ and $x=\lfloor \frac{x}{2}\rfloor$ to $x=\lfloor \frac{x}{2^{k_{s}\left(x\right)}}\rfloor$. Therefore, the pseudo-code of the tANS encoding function becomes: $$tANS\;Encoding\;\{\;k=k_{s}\left(x\right)=\log_{2}\lfloor \frac{x}{L_{s}}\rfloor \;put\;mod\left(x,\;2^{k}\right)\;to\;the\;LSB\;of\;the\;’\left(ANS-\right)\;bitstream\;variable’;\;x=\lfloor \frac{x}{2^{k_{s}\left(x\right)}}\rfloor ;\;x=C\left(s,\;x\right)\;\}$$
Similarly, the pseudo-code of the tANS decoding function becomes:$$tANS\;Decoding\;:\{\;\left(s,\;x\right)=D\left(x\right)\;use\;s;\;k=k\left(x\right)=R-\lfloor \log_{2}\left(x\right)\rfloor \;x=2^{k_{s}\left(x\right)}x+extract\;1\;bit\;from\;the\;MSB\;of\;the\;‘bitstream\;variable’;\;\;\}$$

(b)The construction of Coding Tables for tANS

Based on the discussions above, when the current input state is *x*, the symbol to be encoded is *s*, and the output next state is *x*’, we have $x^{\prime }=C\left(s,\;\frac{x}{2^{k}}\right)$ and the generated bit sequence = $mod\left(x,\;2^{k}\right)$. Therefore, Table 5 and Table 6 illustrate the forms of tANS encoding and the decoding tables, respectively. 

(c)The Complete Encoding and Decoding Processes of tANS

For a symbol sequence to be encoded, tANS starts its encoding from the last symbol of the symbol sequence, then the second to last. The generated bit sequence is storing on the LSB of the bitstream variable during the encoding process. When completing the encoding, a state and a bitstream will be generated. In the opposite direction, tANS starts its decoding with a state and a bitstream. As pre-described, the tANS decoder starts extracting bits from the MSB of the bitstream variable during the decoding.

In short, we can summarize the whole tANS coding process by the following four steps:

Step 1:Calculate the actual symbol probability $p_{s}$ in the to-be-compressed file,Determine the allowable state range I :={L,L+1,…,2L−1} and the state range of each symbol $I_{s}\mathrm{:=}\left\{L_{s},L_{s}+1,\;\ldots ,2L_{s}-1\;\right\},$Use $q_{s}=\frac{L_{s}}{L}$ to approximate $p_{s}$.

Step 2: Determine the proper SSF, $\bar{s}\left(x\right)=s,\;s\;:L\to A$,Establish the tabularized encoding state function composed of $C\left(x,\;s\right)=x^{\prime }$ and a tabularized decoding function composed of $D\left(x^{\prime }\right)=\left(s,\;x\right)$.

Step 3: Determine the encoding and decoding tables according to the SSF determined in Step 2.

Step 4: Start the encoding and decoding processes.

To give readers a clear picture of the operations of tANS encoding and decoding and maintain fluent readability, a concrete and step-by-step example that illustrates the complete tANS processes is given in Appendix C.

### 3.4. The Avalanche Effect of the tANS

As mentioned earlier, tANS encoding processes can be treated as state transitions in a Finite State Machine model. Therefore, as long as the encoding input symbol is different, the encoder will produce (or the model will jump to) different output states even for the same initial state. Under the same condition given in Example C-1, assume there are two different inputs: input one is with the symbol sequence “*cabcaada*,” while input two is with the symbol sequence “*cbbcaada*”. That is, the two inputs are different only at the second symbol. Following the tANS encoding procedure, it is easy to verify that the output corresponds to input one is (State = 16, bitstream variable = “1101111110111111100”), and the result associated with input two is (State = 16, bitstream variable = “110111100010000000”). Notice that the output states are identical, but the bitstreams in the stream variables are dissimilar starting from the second bit, which is where the two input symbol sequences begin to have a difference. In the opposite direction, in tANS decoding, the output states generated during encoding will be used as the starting states of the decoder, and the bitstream stored on the bitstream variable will be extracted to conduct the renormalization process. Because of this mutual chaining nature, as long as the operand state or the content of the bitstream variable is different, the decoded result will be completely different, also.

Like arithmetic coding, this kind of functional behavior that a tiny change in inputs will produce a significant difference in outputs is one of the preferred features called the avalanche effect in cryptography. As mentioned above, the avalanching characteristics of tANS make it applicable to data security protection besides its original well-known usage in data compression.

As for the chaotic behavior of ANS, there is one more thing that is worthy of notice. As addressed in Section 3.2(b) and Section 3.3, the ANS encoding and decoding functions are highly related to the designated SSF, where enormous possible choices exist. In other words, the combinatorial complexity in selecting SSF will lead to a higher degree of chaos, especially for tANS.

## 4. Applications of the Asymmetric Numeral Systems

To provide strong evidence of the value of ANS in practical applications, we review some collected successful and meaningful applications of ANS that have been addressed in the literature so far in this section. Additionally, as a new contribution, the application of ANS to Intellectual Property Rights Management and Integrity Checking of Digital Images will be discussed in detail in Section 4.3.

### 4.1. ANS in Index Compression and Machine Learning-Based Lossless Data Compression

Alistair Moffat and Matthias Petri [22] considered how ANS coding could be used with existing index compression techniques. They showed that ANS could be usefully combined with several index compression approaches to yield improved compression effectiveness within reasonable additional resource costs. By joining ANS with each of byte-based codes, word-based codes, and packed codes, they established new trade-offs for effectiveness and efficiency in index compression. Experiments on an inverted index for the 426 GiB Gov2 collection, the authors showed in [22] that the combination of blocking and ANS-based entropy-coding against a set of 16 magnitude-based probability models yields compression effectiveness superior to most previous mechanisms while still providing reasonable decoding speed. Later, the same authors extended their study to examine the task of block-based inverted index compression [23], in which fixed-length blocks of postings data are compressed independently of each other. Instead of using one parameter, [23] proposed using a two-dimensional selector to summarize each block’s distribution of values. Ref. [23] also introduced a revised mapping from symbol identifiers to ANS values requiring less memory and providing byte-friendly output for exception values. Experiments with two extensive document collections demonstrate that the proposed mechanism can achieve substantial compression gain, and the query throughput speeds are relatively unaffected.

The field of machine learning has experienced an explosion of activity in recent years. We have seen many papers looking at applications of modern deep learning methods, such as AutoEncoder-based and GAN-like mechanisms, to lossy compression. Comparatively, applying Deep Neural Networks (DNNs) to lossless compression has been less well covered in recent works. Ref. [28] seeks to advance in this direction, focusing on lossless compression using latent variable models. In contrast to implementing bits-back coding [37] by Arithmetic codes, ref. [28] suggested using ANS instead and termed the new coding scheme ‘Bits Back with ANS’ (BB-ANS). After conducting a series of experiments, ref. [28] found that BB-ANS with a Variational AutoEncoder (VAE) outperforms generic lossless compression algorithms for binarized and raw MNIST, even with a straightforward one model architecture. The authors of [28] extrapolate these results to predict that state-of-the-art latent variable models could be used in conjunction with BB-ANS to achieve significantly better lossless compression rates than current methods. However, as pointed out by [29], BB-ANS incurs an overhead that grows with the number of latent variables, restricting the capacity of VAE and posing difficulties for density estimation performance; hence, the resulting compression rate suffers. Ref. [29] suggested recursively applying bits-back coding and termed the resulting scheme ‘Bit-Swap’ approach to conquering this shortage. Bit-Swap [29] improves BB-ANS’s performance on hierarchical latent variable models with Markov chain structure. Compared to latent variables models with only one latent layer, these hierarchical latent variable models allow us to achieve better density estimation performance on complex high-dimensional distributions. Although connecting ANS with DNN is out of the focus of this writeup, we do think this is one of the future research directions worthy of further exploration and investigation.

### 4.2. ANS in Joint Compression and Encryption of Digital Images

As a variation of entropy codes, Duda mentioned in his earliest works [2,3] that there is considerable freedom while choosing a specific implementation table for ANS; therefore, we can simultaneously apply ANS to compress and encrypt a message. Duda and Niemiec continue to discuss the applicability of ANS for compression with encryption in [19], pointing out that ANS makes it possible to encrypt the encoded message at nearly no additional cost simultaneously. Moreover, ref. [19] analyzed the security level provided by ANS-based cipher. The main security feature provided by ANS is the pre-described Avalanche effect which comes from ANS’s variable length coding nature. Any attempt to recover from ANS-coded bits to the original symbols has to resolve the error propagation problem caused by even a single bit of erroneous decoding. It is well known that the probability of getting a successful frame synchronization is negligible even for short sequences of symbols and decreases exponentially with the number of compressed symbols. However, as analyzed in [34], plain ANS could only support applications with low-level security requirements. In the same writeup, Seyit Camtepe et al. investigated the natural properties of ANS, allowing incorporation with authenticated encryption using as little cryptography as possible. Moreover, they proposed three joint compression and encryption algorithms to face real applications with much higher security requirements. The first applies a single ANS with state jumps controlled by a pseudorandom bit generator (PRBG). The second one uses two copies of ANS, where PRBG manages the transition between the two ANSs. The third algorithm deploys encoding function evolution to enhance the obtained security level. The contributions of [34] boomed up the applicability of ANS in joint compression and encryption a lot. 

As mentioned in [34], though, the randomness of the pure Avalanche effect-based encryption scheme is not enough to deal with high-level security applications. There are cases where low-level security may be workable with the aid of other control mechanisms. For example, the distribution of art collections and the verification of medical images are under particular management rules, which is quite different from the communications scenario among IoT sensors or devices considered in [34]. We believe that ANS might still provide a useful jointly compressing and encrypting function for those applications. Therefore, we will investigate the possibility of applying ANS to protect the intellectual property rights (IPRs) of art collection pictures or check the integrity of medical images in the next section.

### 4.3. ANS in Intellectual Property Rights Management and Integrity Checking of Digital Images

To exactly recover a time signal from its frequency domain representation, we need to know both the magnitude and phase responses of the signal. Likewise, in ANS, both the correct state value and content of the bitstream variable are a must for reconstructing a digital image without loss. Based on its avalanche effect, we can apply tANS as a vehicle to protect the intellectual property rights (IPRs) of art collection pictures or check the integrity of medical images as described in the following sub-sections.

(a)Some Specific Characteristics of ANS

Before going into the details, let us recall several preferred features provided by ANS.

1.Lossless and Compressive Representation

As pre-described, ANS belongs to the category of entropy coding; lossless compression is undoubtedly one of its profound properties. Therefore, it is pretty suitable for being applied to digital art collection images or medical images, where compact and distortion-free representation is of top priority. 

Moreover, ANS provides a compression efficiency close to the Shannon limit, but relatively few researches of ANS on image compression exist. The JPEG Standard committee proposed JPEG XL [8] in 2017, in which the entropy coder changed to use rANS. Since JPEG XL includes many pre-processing and optimization techniques, its reported compression efficiency is better than the naive approach adopted in this work.

2.Avalanche and Retrospective Properties

The avalanche effect mentioned above is quite suitable for providing a compact representation of digital art collection images. We can represent a digital art image by a positive integer state and a bit sequence. Art collectors can open, says the state, to the public as the evidence for claiming the ownership of this artwork and keep the bit sequence in private as the verifier if a dispute occurs. Because of its retrospective and avalanche characteristics, we think there will be an excellent opportunity to combine ANS with the recently popular NFT (Non-Fungible Token) [11] to make the IPRs of artwork much more secured. Similarly, we can use these two properties to check the integrity and protect the privacy of medical images at the same time.

3.Severability

We can apply the compactness and the lossless properties of ANS mentioned above to digital images in a block-segmented way. With ANS’s segmentable feature, we can assign different levels of protection or degrees of integrity checking to various portions of an image according to their importance. An artwork publisher who intends to sell his digital artworks to more than one artwork collector can divide his art collection into different pieces and price them according to the corresponding values. Then, the publisher can generate the state and the bit sequence for representing each partition. He can now disclose the state information to the potential customers as a marketing representative of this partition in NFT applications. Moreover, the bit sequence of the same segmented area can then be sent to the actual buyer as a voucher for certifying the ownership. Moreover, from the marketing point of view, through the integration of ANS and NFT, a single physical artwork collection can be distributed, shared, and sold in the virtual world, which enlarges the potential market size and magnifies the market value of a digital artwork substantially.

(b)The Proposed Applications of ANS-based Digital Image Processing System

Figure 3 shows the information flow of the proposed ANS-based digital image processing system. A bank of ANS encoders is used to encode a given image, where each encoder generates a state and a bitstream representation for a given portion of the segmented input image. All the generated state values are collected to form a state-map of the image, which is made public and openly distributed in our system as a digital representation of that particular picture. On the contrary, we keep the collection of generated bitstreams in the artist’s (or a museum official’s) hands as proof of the ownership of that image (i.e., the digital artwork). Notice that we include a segmentation mask into our system, indicating the geometric pattern and the number of portions the input image could be partitioned. With the aid of the mask, we can process different portions of an image with distinct ANS encoders, where different SSFs are adopted to offer various realizations of ANS coding functions. The more complex and erratic the mask is the higher our system’s security protection.

Figure 4 shows the actual encoder we used to enhance our system’s security protection capability. We separate the input image into RGB components and segment each color component into equal-sized blocks (called them sub-images) simply for ease of implementation. Additionally, we add a block-based shuffling module to our system to increase the confusion ability of our system. Finally, Figure 5 shows the block diagram of the actual decoder used in our system. Of course, we can treat the key used to conduct the block-based permutation as one of the security parameters of the proposed system.

## 5. Experimental Results

Through a series of experiments, we examine the applicability of the proposed tANS-based system to protect IPRs of digital artwork collections and the integrity of medical images in this section. The following experiment is conducted in Darwin MacBook-Pro.local 18.7.0 Darwin Kernel Version 18.7.0; root:xnu-4903.278.44~1/RELEASE_X86_64 x86_64 computer system. For the ANS algorithm, we choose new generation entropy codecs: Finite State Entropy from [38], which is the first implementation of ANS developed by Yann Collet.

### 5.1. tANS in IPRs Protection of Digital Artwork Collections

This section will utilize the segmentable and retrospective features of tANS to protect the IPRs of an artwork image. To make readers better understand what we are doing, let us examine the related processing flow for the digitized painting picture shown in Figure 6. (We choose a low-resolution picture as the testing benchmark to avoid violating copyrights. ANS coding operations will not affect the processed image quality because they are conducted in the integer domain.)

Different colors in the mask define geometric patterns for different segmenting sub-images according to various degrees of importance about the image’s content. As previously addressed, a tANS encodes a sub-image into an outputs state and an associated bitstream. As shown in Figure 7, our first experiment is to change one byte of the state value in the encoded domain to see whether the decoded result will show the so-called avalanche effect.

We randomly pick a sub-image defined by one specific color in the mask. Then, we randomly change a byte of the state value of the chosen sub-image. We observe the corresponding decoded output for the following two things:(1)Does the damaged area of the decompressed image locate in the same areas where the state value changed?(2)Is the degree of contamination in the damage severe or not?

There are ten distinct areas with different geometric patterns defined in the mask in our experiments. Figure 8 shows the snapshots corresponding to each sub-images, where one byte of the state value in each sub-image is changed randomly.

We measure our experiment’s compression performance based on the compression ratio, defined as the file size before compression to the file size after compression. The average compression ratio of our experiments is 88%. This ratio is not very impressive as compared with conventional entropy coders. The reason behind this not-so-good compression performance is that we did not take many pre-processing and optimization techniques into account, which have been proved effective in enhancing compression performance in JPEG XL. Another possible factor for the not-so-impressive compression performance comes from the usage of tANS. Although tANS is one branch of ANSs, which provides the best efficiency in realization and processing speed, it is not optimized for image compression. This fact tells us that still there is a large room for us to develop ANS-based approaches for providing good performance both in security protection and compression ratio. As for the degree of contamination, those completely black blocks in the sub-images of the decompressed picture tell us that the avalanche effect causes 100% of the impact, even though only a one-byte state value is changed. 

### 5.2. tANS in Integrity Checking of Digital Medical Images

Enforcing protection of the contents of medical imaging, such as computed tomography (CT), magnetic resonance (MR), positron emission tomography (PET), mammography, ultrasound, X-ray, has become a significant issue of computer security. Except for their being valuable and essential for the early detection, diagnosis, and treatment of diseases, their more and more widespread distribution makes developing security mechanisms to guarantee their confidentiality, integrity, and traceability in an autonomous way becomes a must. Facing such a demand, researchers proposed Reversible Watermarking (RW) [39,40] schemes for images of sensitive content, e.g., medical images, such that any modification may aspect their interpretation. However, extra data (the watermark) must be embedded in the protection target, which usually increases the file size. In this section, we suggest using tANS as the representative of the medical image content to achieve medical images’ security protection and file size reduction simultaneously.

We use the same system given in Figure 7 to test the integrity of medical images. Figure 9 and Figure 10, respectively, show the original input and contaminated output images. Notice that the ability to check Images’ integrity comes from tANS’ avalanche feature while the segmentability of tANS contributes to parallelizability in implementation.

## 6. Performance Comparison among Various Lossless Compression Algorithms

As suggested by anonymous reviewers, the comparisons of the performance among various lossless compression algorithms, in terms of compression ratio and execution speed, are reported in this section. 

### 6.1. Description of Experimental Settings

Environment setting is Darwin MacBook-Pro.local 18.7.0 Darwin Kernel Version 18.7.0; root: xnu-4903.278.44~1/RELEASE_X86_64 x86_64; the language is python; the editor is jupyter notebook. The algorithms we took for comparison included Huffman coding, Arithmetic Coding, lzma, zlib, gzip, bz2, rANS, Finite State Entropy(tANS), zstd(tANS), and liblzfse(tANS) with sources from [41,42,43,44,45,46,47,48,49,50,51,52,53]. In particular, zstd and liblzfse are the two table ANS-based algorithms respectively proposed by Facebook and Apple. Our choice with these algorithms is to compare different lossless compression algorithms against the ANS counterpart.

The pictures we chose for benchmarking include all black, lattice, Lena, fruits, baboon, airplane, and chest images with sources from [54]. The reason we choose these images is for diversity. Our choice with the all-black and the black-and-white lattice images is to see how the algorithms, as mentioned above, performed on low entropy images with one color and two colors. Similarly, our choice with Lena, fruits, baboon, and airplane images is to see how those algorithms performed on classic gray images used in image processing communities. Finally, we chose the chest image is to see how these algorithms performed on a medical image. In order to show how different these pictures are, we show their histograms also. By the way, our experiments did not involve any preprocessing of the testing images; therefore, the compression ratios do not as good as expected. However, we can still see the performance differences among all these algorithms. 

### 6.2. Experiment Results

In the following, we will take the all-black image as an illustrative example to address and explain our experiment’s procedures and results first. The leftmost (a) and the middle (b) pictures of Figure 11, respectively, show the all-black image’s snapshot and histogram, while the rightmost (c) chart reports the compression ratios of all algorithms we want to compare. Here, the compression ratio is defined as the ratio of the compressed file size to the uncompressed one. The x-axis of the middle picture denotes the image’s RGB value, which ranges from 0 (pure black) to 255 (pure white); the corresponding y-axis represents the number of appearances of each RGB value. Histograms help characterize how different colors are distributed within an image. Notice that the higher the height of the bar is in the (c) chart, the poorer the compression performance.

Following the same arguments, Figure 12, Figure 13, Figure 14, Figure 15, Figure 16 and Figure 17 show the related experimental information associated with the black-and-white lattice, the Lena, the fruits, the baboon, the airplane, and the chest images, respectively.

For providing a clear picture of the relative compression ratio comparison, Table 7 shows the compression ratio of each algorithm in numerals. Moreover, to show the timing performance of all benchmark algorithms, Table 8 reports the time consuming of each tested algorithm we obtained in seconds. Notice that, as abovementioned, no optimization preprocess has been included in any of our experiments.

### 6.3. Observations Obtained from Our Experiments

Those rows with gray backgrounds in Table 7 report compression ratios obtained from ANS-related algorithms. We could make some comments to these results:ANS-related algorithms performed well if the data distribution is highly skewed, inferred from the all-black and the lattice images.ANS-related algorithms performed generally if the data distribution is almost uniform, inferred from Lena, fruits, and baboon images.ANS-related algorithms performed well for medical images (c.f., chest image) because most of the area in a medical image is of the same color black or white), which also coincides with our first comment.As for the compression ratio, we found that ANS-related algorithms performed almost the same as the arithmetic coding or a little bit better, which is as expected from the theoretical point of view.As for the time consumption, we found that ANS-related algorithms almost need the least execution time among all algorithms and are comparable to the Huffman code, which is also as expected from the theoretical point of view.

As we said in Section 6.2, we did not employ any image preprocessing before compressing the images. This fact explains why the compression ratios of some images are not as good as expected. In general, some steps before applying the entropy coding are a must within a standard image compression algorithm. For example, in pik, which Google releases and adopts ANS as its source coding component, it involved some image preprocessing techniques to enhance its compression performance.

## 7. Conclusions

ANS is valued by the industry precisely because it captures the benefits of both Huffman coding and arithmetic coding. Surprisingly, compared with Huffman and arithmetic coding, the application of ANS to image compression is rare. Therefore, this paper intends to give a self-contained review of ANS-related technologies in depth and apply them to compress and encrypt digital images. ANS’s lossless compression feature makes it especially suitable for distortion-less compression-related applications, such as medical and digital art collection images. The retrospective of ANS comes from its avalanche breakdown characteristic, which can easily be realized by using table ANS. Further, we suggested combining ANS with the recently popular NFT (non-fungible token) to make the intellectual rights of artwork much more secure.

In addition, as application examples, we explored the feasibility of using ANS to art collection images and medical images. We thoroughly investigated ANS’s avalanche effect, which makes ANS applicable to lossless compression, segmentation, and retrospective digital images. Moreover, we successfully applied ANS’s avalanche characteristic and segmentability to check the integrity of medical images in parallel.

As ANS is still under development, there is enormous room for future research. We list some topics that we plan to explore shortly in the following:(1)The combinatorial complexity in designing proper SSF makes developing an optimal ANS codec concerning a specific target becomes very challenging. Thus, finding a heuristic approach for reaching an effective ANS solution for a given input source is of great interest.(2)Based on the obtained states and bitstreams, develop some post-processing, such as prefix or suffix coding, or go through a hash function to find a unique state representation is worthy of doing.(3)Develop an efficient way to combine image recognition and segmentation techniques to automatically find Region of Interests (ROIs) in a picture so that the mask does not need to be manually set. This subject is of interest and beneficial to those planning to develop ANS-based image protection applications systematically and automatically.(4)Since one of the tANS coded results is a bitstream, which indeed can be losslessly compressed again to make the space smaller, then, “what is the best combination of all possible entropy coders?” would be an exciting research topic.(5)Before the image enters the ANS coder, it can be processed (transformed) in advance. Since the mask can divide an image at will, applying other image processing techniques to a sub-image with arbitrary shapes becomes challenging.(6)As mentioned at the end of Section 4.1, properly combining ANS with DNN to produce a fast compression mechanism with a high compression ratio is a research direction worthy of further exploration and investigation.

## Figures and Tables

**Figure 1 entropy-24-00375-f001:**
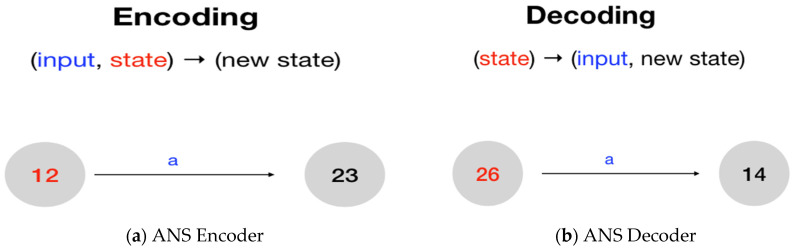
ANS coding in State Transition Form: (**a**) ANS Encoder and (**b**) ANS Decoder.

**Figure 2 entropy-24-00375-f002:**
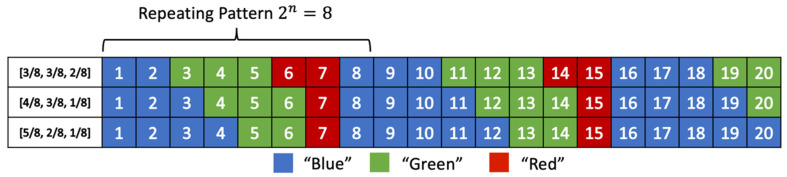
The state arrangements in the rANS coding table for various probability distributions of the three symbols in Example 3, where the Blue-block is used to denote symbol a, Green-block is for b, and Red-block is for c.

**Figure 3 entropy-24-00375-f003:**
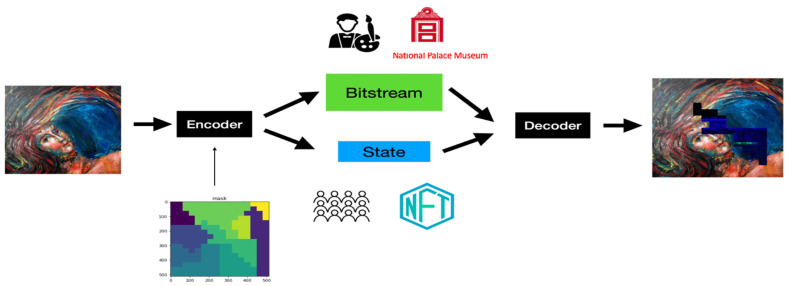
Block diagram of the proposed ANS-based digital image processing system for IPR protection of digital artwork collections. (Notice that the output image of the above figure has been slightly enlarged to show the effect of segmented masking.).

**Figure 4 entropy-24-00375-f004:**
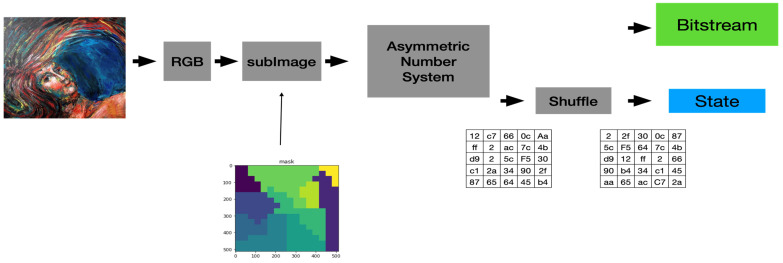
The Block Diagram of the Actual Encoder Adopted in Our System.

**Figure 5 entropy-24-00375-f005:**
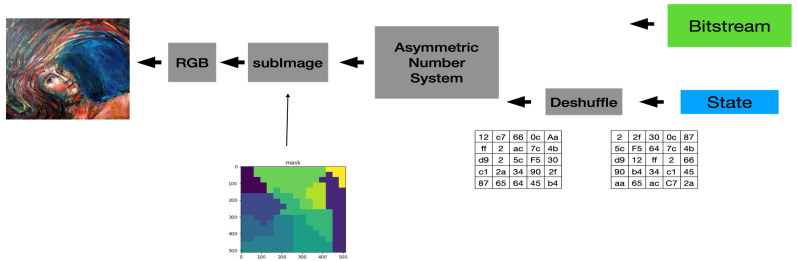
The Block Diagram of the Actual Decoder Adopted in Our System.

**Figure 6 entropy-24-00375-f006:**
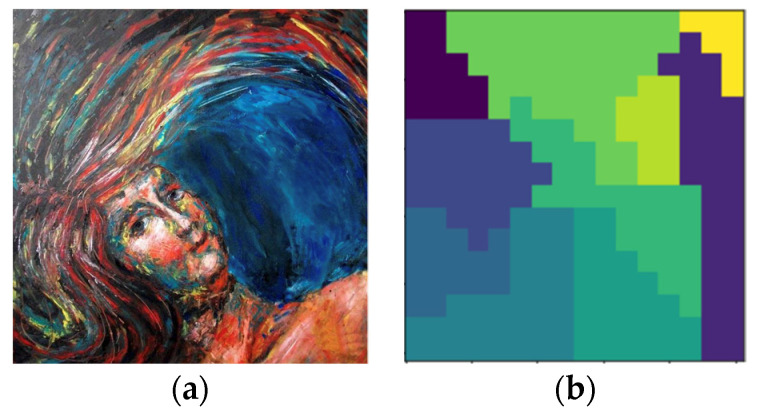
(a) The painting picture to be protected and (b) the mask used to segment the picture in (a).

**Figure 7 entropy-24-00375-f007:**
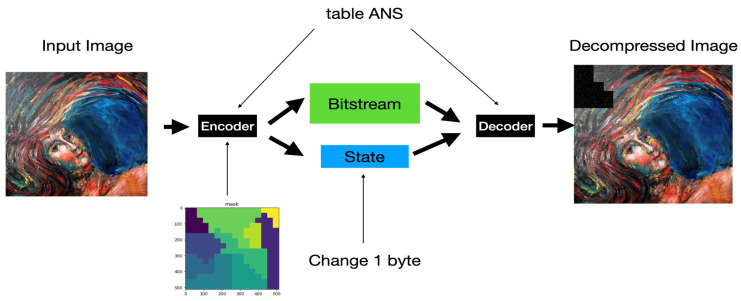
Flow chart of experiments for ANS’s avalanche effect in one-byte state value change. (Notice that the input, the output, and the mask images are of the same size.).

**Figure 8 entropy-24-00375-f008:**
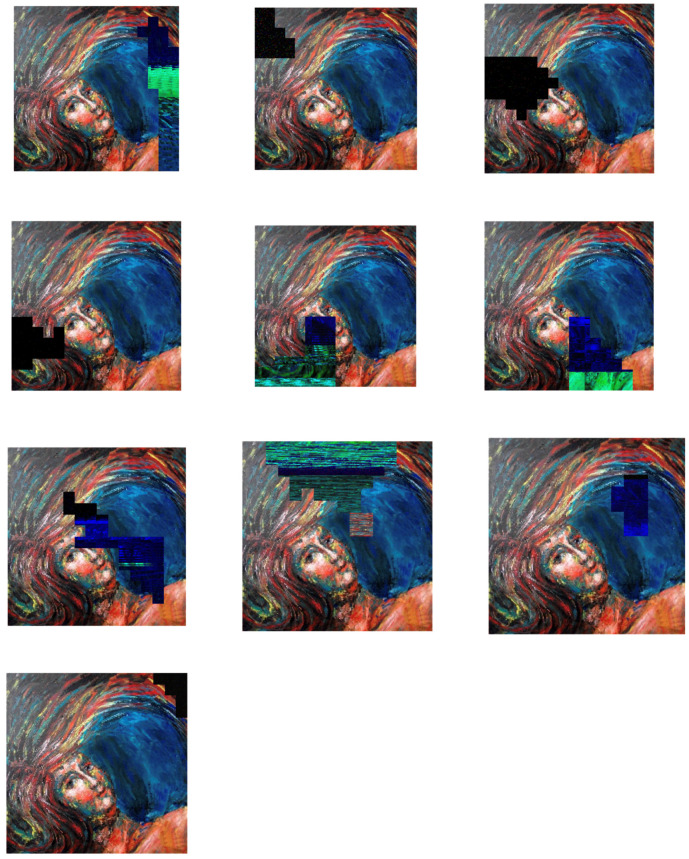
Snapshots of each sub-images, where one byte of the state value in each sub-image is changed randomly. (Notice that the byte change is zero in the most bottom sub-image, i.e., it is the original image.)

**Figure 9 entropy-24-00375-f009:**
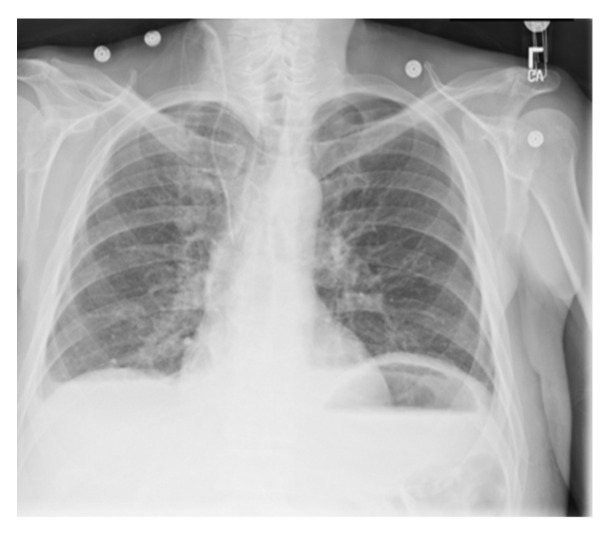
The original testing medical image.

**Figure 10 entropy-24-00375-f010:**
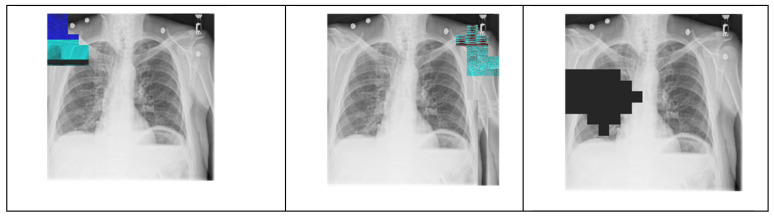
Snapshots of each sub-image, where one byte of the state value in each sub-image is changed randomly. (Notice that in the most bottom sub-image, the byte change is zero.) The independence of the contamination of each sub-image shows the ability to conduct the integrity checking of the whole image in parallel.

**Figure 11 entropy-24-00375-f011:**
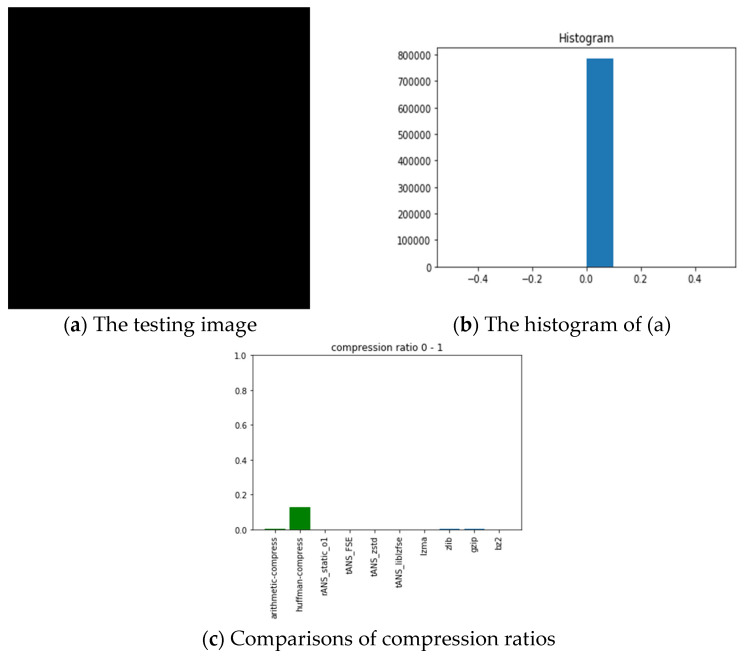
The experimental related information associated with the all-black image.

**Figure 12 entropy-24-00375-f012:**
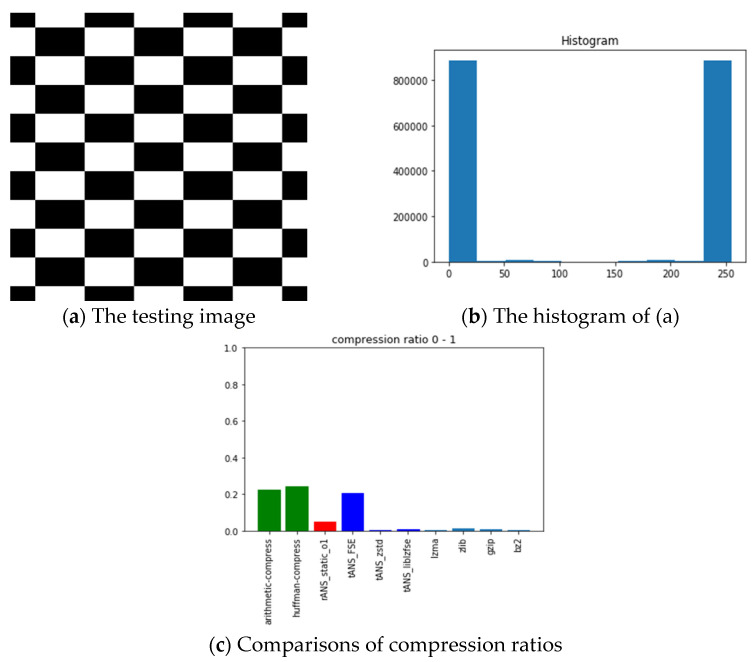
The experimental related information associated with the black-and-white lattice image.

**Figure 13 entropy-24-00375-f013:**
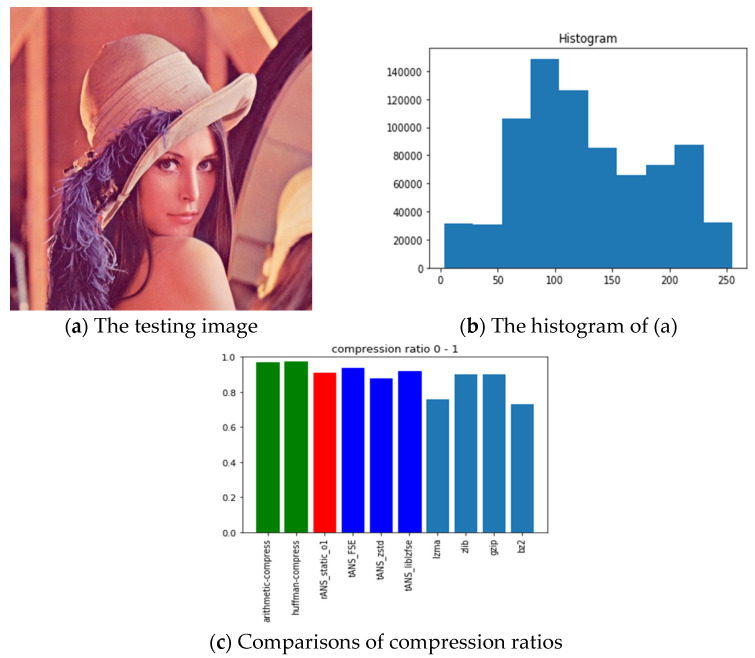
The experimental related information associated with the Lena image.

**Figure 14 entropy-24-00375-f014:**
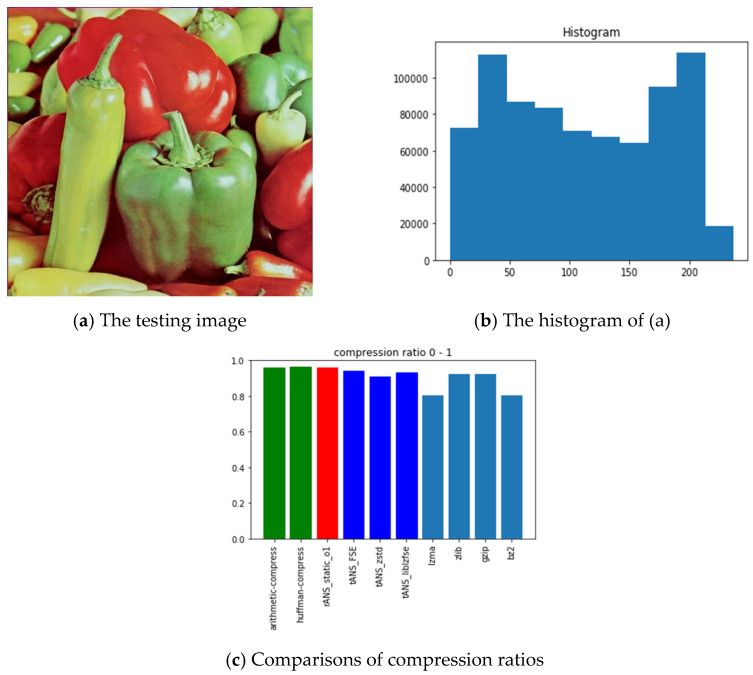
The experimental related information associated with the fruits image.

**Figure 15 entropy-24-00375-f015:**
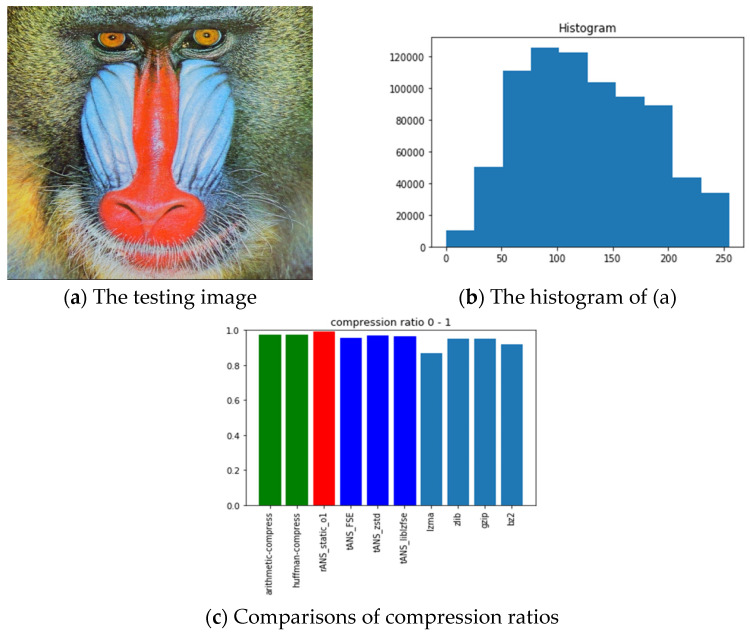
The experimental related information associated with the baboon image.

**Figure 16 entropy-24-00375-f016:**
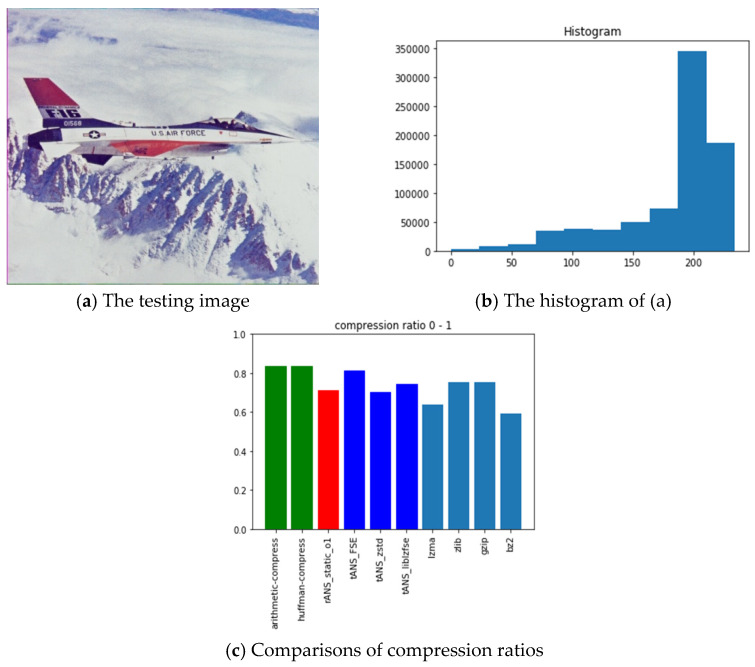
The experimental related information associated with the airplane image.

**Figure 17 entropy-24-00375-f017:**
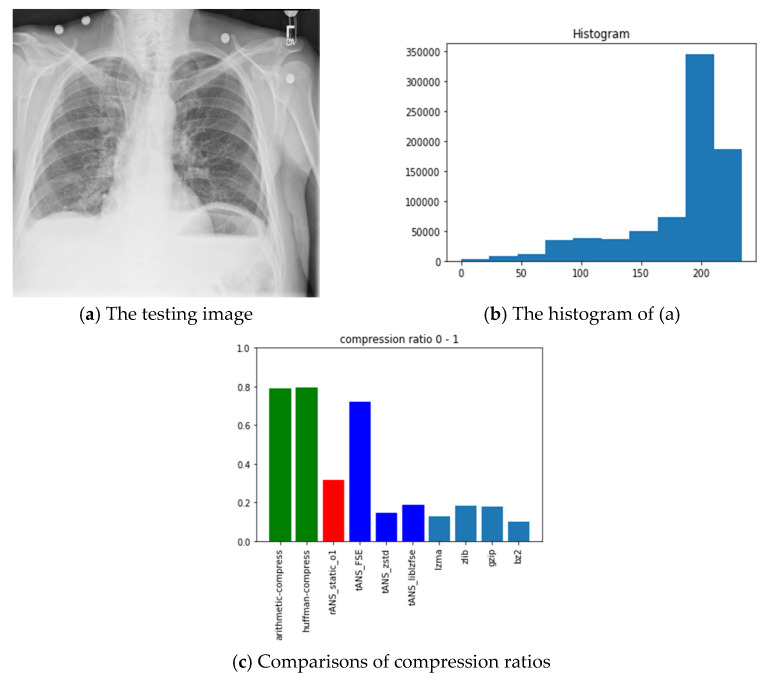
The experimental related information associated with the chest image.

**Table 1 entropy-24-00375-t001:** The uABS’s Encoding and Decoding in the Form of a Coding Table.

x′		0	1	2	3	4	5	6	7	8	9	10	11	12	13
*x*	*s* = 0	0		1		2		3		4		5		6	
*x*	*s* = 1		0		1		2		3		4		5		6

**Table 2 entropy-24-00375-t002:** The uABS’s Coding Table for Input “01111”.

*x*′		0	1	2	3	4	5	6	7	8	9	10
*x*	*s* = 0	0					1					2
*x*	*s* = 1		0	1	2	3		4	5	6	7	

**Table 3 entropy-24-00375-t003:** The rANS Coding Table Realization for Example 3.

x′		0	1	2	3	4	5	6	7	8	9	10	11	12	13	14	15	16	17	18	19	20	21	22	23
x	s = a	0	1	2	3	4				5	6	7	8	9				10	11	12	13	14			
x	s = b						0	1							2	3							4	5	
x	s = c								0								1								2

**Table 4 entropy-24-00375-t004:**
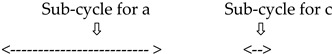
The Schematic Meaning of Sub-cycles of Symbols, for an rANS Coding Table, in Example 3.

x′		0	1	2	3	4	5	6	7	8	9	10	11	12	13	14	15
x	s = a	0	1	2	3	4				5	6	7	8	9			
x	s = b						0	1							2	3	
x	s = c								0								1

**Table 5 entropy-24-00375-t005:** The General Form of tANS Encoding Table.

		*x* = *L*	*x* = *L + 1*	…	*x* = *2L − 1*
*s*					
…					
*s_i_*			next state = $C\left(\mathrm{si},\;\lfloor \frac{x}{2^{k}}\rfloor \right)$; bit sequence = $mod\left(x,\;2^{k}\right)$		
…					
*s_n_*					

**Table 6 entropy-24-00375-t006:** The General Form of tANS Decoding Table.

*x*′ (Current State)	*L*	*L + 1*	*…*	*2L − 1*
*s* (generated symbol)				
*K* (# of bits extracted from the bitstream Variable)				
*X* (next state)				

**Table 7 entropy-24-00375-t007:** Compression ratios of all benchmark images in all tested algorithms.

	Picture	All Black	Lattice	Lena	Fruits	Baboon	Airplane	Chest
Algorithm								
Arithmetic-code		0.00131	0.22174	0.97008	0.96003	0.97161	0.83429	0.79038
Huffman-code		0.12533	0.24435	0.97291	0.96212	0.97544	0.83673	0.79266
rANS		4 × 10^−5^	0.047	0.90697	0.95826	0.98997	0.71251	0.31487
tANS_FSE		7 × 10^−5^	0.2058	0.93765	0.94194	0.95575	0.81241	0.72081
tANS_zstd		5 × 10^−5^	0.00238	0.87594	0.90971	0.96825	0.70071	0.14375
tANS_liblzfse		0.00077	0.00576	0.91833	0.93128	0.96233	0.74537	0.18732
lzma		0.00031	0.00214	0.75684	0.80398	0.86879	0.63955	0.12609
zlib		0.001	0.0132	0.89948	0.92114	0.94867	0.75443	0.18086
gzip		0.00101	0.00967	0.89949	0.92116	0.94869	0.75447	0.17982
bz2		6 × 10^−5^	0.00216	0.73081	0.80389	0.91668	0.59302	0.1007

**Table 8 entropy-24-00375-t008:** Time consuming of all benchmark image in all tested algorithms (seconds).

	Picture	All Black	Lattice	Lena	Fruits	Baboon	Airplane	Chest
Algorithm								
Arithmetic-code		0.00543	0.00416	0.00395	0.00485	0.00646	0.00515	0.0057
Huffman-code		0.00249	0.0022	0.00206	0.00263	0.00247	0.00267	0.00263
rANS		0.00183	0.00223	0.00178	0.002	0.00233	0.00196	0.00228
tANS_FSE		0.00241	0.00212	0.00189	0.00219	0.0018	0.00188	0.0017
tANS_zstd		0.01235	0.11686	0.11693	0.12242	0.10236	0.18194	1.84513
tANS_liblzfse		0.00887	0.0187	0.01367	0.01394	0.01223	0.01776	0.07471
lzma		0.03119	0.15142	0.19254	0.19741	0.18212	0.20994	2.00818
zlib		0.00372	0.01307	0.02393	0.02478	0.02246	0.02851	0.15271
gzip		0.00449	0.0497	0.024	0.02522	0.02261	0.02967	0.2924
bz2		0.00866	0.07017	0.06489	0.07004	0.07578	0.06919	0.19868

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
