# Peer review of "A Review of the Asymmetric Numeral System and Its Applications to Digital Images"

_entropy, 2022, doi:10.3390/e24030375_

Round 1
Reviewer 1 Report
This paper is dedicated to joint compression and encryption of images using entropy coding based on Asymmetric Numeral System (ANS). In general, I think this is a good overview of the ANS concept, but some major parts of the manuscript should be modified.
Comments to the authors:
- I think, the authors should write the article in more math format. For example, usage of “-->” is not acceptable. The same is related to font style of math symbols within the text, for example at lines 430-438 and many more.
- The quality of pictures is too low.
- The authors state “The first one is that the corresponding compressed output will change a lot if there is a tiny change in the original input.” But I think, the same is true for arithmetic coding as well.
- It could be nice of the authors provide some numbers, showing the comparison in encoding/decoding speed for Huffman coding, AC and ANS, as well as its compression efficiencies.
- The presented description of arithmetic coding (AC) is actually wrong, since modern implementations of arithmetic coding use registers renormalization which helps to avoid floating point operations. First such fully integer multi-symbol implementation of AC was proposed in 1987 in [a]. But implementation in [a] has multiplications and divisions, therefore it was proposed several look-up table based adaptive binary arithmetic coding implementations, for example, the most famous among them is M-coder [b] which is the core of entropy coding in H.264/AVC, H.265/HEVC, H.266/VVC video coding standards, and MQ-coder [c] which are used in JPEG2000 image coding standard. Moreover, there are more advanced research work related to adaptive range coding (AC with fast renormalization), for example [d], and multiplicaition and division free multi-symbol AC [e]. I think, the authors should rewrite Section 2.2, i.e., add references [a]-[e], and write that modern AC are not floating point algorithms.
[a] I. Witten, R. Neal, and J. Cleary, “Arithmetic coding for data compression, “Communications of the ACM, vol. 30, pp. 520–540, 1987.
[b] D. Marpe, H. Schwarz and T. Wiegand, “Context-based Adaptive Binary Arithmetic Coding in the H.264/AVC Video Compression Standard, “IEEE Transactions on Circuits and Systems for Video Technology, vol.13, no.7, pp.620–636, 2003.
[c] ITU-T and ISO/IEC JTC 1, “JPEG 2000 image coding system: Core coding system, ITU-T Recommendation T. 800 and ISO/IEC 15444-1,“ JPEG 2000 Part 1, 2000.
[d] E. Belyaev, K. Liu, M. Gabbouj, and Y. Li, “An efficient adaptive binary range coder and its vlsi architecture,” IEEE Transactions on Circuits and Systems for Video Technology, vol. 25, no. 8, pp. 1435–1446, 2015.
[e] E. Belyaev, S. Forchhammer and K. Liu, "An Adaptive Multialphabet Arithmetic Coding Based on Generalized Virtual Sliding Window," in IEEE Signal Processing Letters, vol. 24, no. 7, pp. 1034-1038, July 2017, doi: 10.1109/LSP.2017.2705250.
- I think, in the overview the authors also need to mention that ANS based coding is not adaptive, i.e., probability of a symbols is not estimated as it is performed in adaptive AC.
- I think, it could be nice to add coding example for 01111 to Section 3.1 (b) as well.
- Figures 2-1 and 2-2 should be improved, because now some of blocks are not connected with others, some tables are given without labels.
Author Response
First, the authors would like to express their deep appreciation to the anonymous reviewer. Without the reviewer's valuable comments, the quality of this work cannot be polished as it is now!
Point-to-Point Reply to Reviewer -1 Comments:
Comments and Suggestions for Authors:
This paper is dedicated to joint compression and encryption of images using entropy coding based on Asymmetric Numeral System (ANS). In general, I think this is a good overview of the ANS concept, but some major parts of the manuscript should be modified.
Our reply:
a. As pointed out by the reviewer, one part of our contributions in this work is providing a step-by-step review of the construction for various versions of ANS. To emphasize this fact, we change the title of the paper to “A review of the Asymmetric Numeral System and Its Applications to Digital Images.”
b. A paragraph is added at the beginning of Section 4 to point out our contribution other than a review writeup.
Comments to the authors:
1. I think, the authors should write the article in more math format. For example, usage of “-->” is not acceptable. The same is related to font style of math symbols within the text, for example at lines 430-438 and many more.
Our reply:
As suggested, we have modified the above inappropriate usages in the revision.
2. The quality of pictures is too low.
Our reply:
As suggested, we have enhanced the qualities of pictures as much as in the revision. To not violate IPR, the adopted art image for illustration is of low quality; however, this choice does not affect the experimental results because all operations involved in ANS are in the integer domain.
3. The authors state “The first one is that the corresponding compressed output will change a lot if there is a tiny change in the original input.” But I think, the same is true for arithmetic coding as well.
Our reply:
We agreed with this reviewer’s comment and included this comment in Arithmetic Coding related descriptions!
4. It could be nice of the authors provide some numbers, showing the comparison in encoding/decoding speed for Huffman coding, AC and ANS, as well as its compression efficiencies.
Our reply:
Thanks for this valuable comment. We add a new Section (Section 6) in the revision
to report our experimental results for comparing nearly all well-known lossless compression algorithms with benchmark testing images usually used in the image communities.
5. The presented description of arithmetic coding (AC) is actually wrong, since modern implementations of arithmetic coding use registers renormalization which helps to avoid floating point operations. First such fully integer multi-symbol implementation of AC was proposed in 1987 in [a]. But implementation in [a] has multiplications and divisions, therefore it was proposed several look-up table based adaptive binary arithmetic coding implementations, for example, the most famous among them is M-coder [b] which is the core of entropy coding in H.264/AVC, H.265/HEVC, H.266/VVC video coding standards, and MQ-coder [c] which are used in JPEG2000 image coding standard. Moreover, there are more advanced research work related to adaptive range coding (AC with fast renormalization), for example [d], and multiplication and division free multi-symbol AC [e]. I think, the authors should rewrite Section 2.2, i.e., add references [a]-[e], and write that modern AC are not floating-point algorithms.
Our reply:
a. Thanks for the comment about the "modern implementations of arithmetic coding use registers renormalization," which helps avoid floating-point operations.We learned a lot from this valuable comment.
b. As suggested, we rewrote most of the latter half of Section 2.2 and added three new references to the revision.
6. I think, in the overview the authors also need to mention that ANS based coding is not adaptive, i.e., probability of a symbols is not estimated as it is performed in adaptive AC.
Our reply:
We agreed with this reviewer’s valuable comment! We added a paragraph to emphasize this fact and put it as one of our future research topics at the end of Section 7.
7. I think, it could be nice to add coding example for 01111 to Section 3.1 (b) as well.
Our reply:
As suggested, this example with skewed probability distribution has been added in the revision.
8. Figures 2-1 and 2-2 should be improved, because now some of blocks are not connected with others, some tables are given without labels.
Our reply:
As suggested, the quality of the mentioned figures has been enhanced, as possible as we can.
Reviewer 2 Report
The article talks about the application of ANS to digital images.
- The title of the article is not appropriate and therefore should be changed. One suggestion is: Application of Asymmetric Numeral System to Digital Images.
- Abstract is simply providing the importance of ANS. The other critical details such as methodology, contributions, validation, results and significance are not present. A probable organization for the abstract is suggested here:
(a) Describe the problem and its importance (Issues in compression of digital images)
(b) Existing compression practices and identify their limitations
(c) What is ANS and how it can solve the identified limitations
(d) Proposed solution (application of ANS to digital images) and its main features
(e) How the proposed approach has been validated
(f) What are the outcomes
(g) What is the significance of outcomes
- Similar to the abstract, Introduction is also required to reorganize. A probable organization for the abstract is suggested here:
(a) Describe a brief background on compression techniques in digital images and what is really required here (identify the limitations).
(b) Describe a brief background on ANS and how it can solve the identified limitations
(c) Description of Proposed solution (application of ANS to digital images) and its main features
(d) How the proposed approach has been validated (description of case studies or bench marks)
(f) What are the outcomes and what is the significance of outcomes
- From lines 70 to 777, (Section 2, Section 3, Section 4.1, Section 4.2 Section 4.3) contain a lot of background information on ANS. While the understanding of basic working of ANS is important to understand this article, it is really useless to consumes almost 700 (which is almost three fourth (3/4 or 75 percent) of the entire article length. Therefore, it is highly recommended to remove all the unnecessary details. The basic working of ANS and its applications can be summarized in a single section.
- A literature review section (Section 3) is needed to understand the current compression methods in digital images. How ANS is different from the current compression methods.
- Section 4 should describe the Proposed ANS-based Digital Image Processing System. Figure 4.1, 4.2 and 4.3 should be elaborated in more detail.
Section 5 should describe the experimental results. This section is really the place where authors can describe that how they have implemented the solution (ANS to digital images). It may contains the following subsection:
- Experimental setup
- Description of tools and particular settings used in the experiment
- Description of Benchmarks and case study.
- Experimentation Procedure
- Experimental Results: summarize the results in some tabular or graphical form and discuss in details.
- Section 6 should provide the Performance Comparison and Discussion. Compare your results (obtained in the previous section) with state-of-art solutions, using tables, graphs and any other methods for your comparison. Good data visualization techniques may enhance the value of your article. Discuss each figure, table or chart in the text (detailed description). Briefly that what is the purpose and message of each figure or table. Use various performance parameters for comparisons. Each performance parameter must be defined explicitly before its use. You should also provide the justification and motivation behind the selection of each performance parameter. It is VERY IMPORTANT to highlight the strengths of your article as compared to existing methods. Discuss the reasons that why you were able to obtain good results. Similarly, it is equally important to mention the shortcomings of your solution with respect to current methods. Discuss the reasons for these limitations and shortcomings.
Minor Issues:
- Abstract should be in the form of a single paragraph.
- The formatting is not consistent.
- Article should be checked for typos.
Author Response
First, the authors would like to express their deep appreciation to the anonymous reviewer. Without the reviewer's valuable comments, the quality of this work cannot be polished as it is now!
Point-to-Point Reply to Reviewer -2 Comments:
Reviewer 2 Comments:
Comments and Suggestions for Authors
The article talks about the application of ANS to digital images.
- The title of the article is not appropriate and therefore should be changed. One suggestion is: Application of Asymmetric Numeral System to Digital Images.
Our Reply:
Since one part of our contributions in this work is providing a step-by-step review of the construction for various versions of ANS. To emphasize this fact, we change the title of the paper to “A review of the Asymmetric Numeral System and Its Applications to Digital Images.”
- Abstract is simply providing the importance of ANS. The other critical details such as methodology, contributions, validation, results and significance are not present. A probable organization for the abstract is suggested here:
(a) Describe the problem and its importance (Issues in compression of digital images)
(b) Existing compression practices and identify their limitations
(c) What is ANS and how it can solve the identified limitations
(d) Proposed solution (application of ANS to digital images) and its main features
(e) How the proposed approach has been validated
(f) What are the outcomes
(g) What is the significance of outcomes
Our reply:
Thanks for all the above comments, which are valuable to improve the Abstract’s quality of our work; we have rewritten the abstract and considered all the suggestions as much as possible.
- Similar to the abstract, Introduction is also required to reorganize. A probable organization for the abstract is suggested here:
(a) Describe a brief background on compression techniques in digital images and what is really required here (identify the limitations).
(b) Describe a brief background on ANS and how it can solve the identified limitations
(c) Description of Proposed solution (application of ANS to digital images) and its main features
(d) How the proposed approach has been validated (description of case studies or bench marks)
(f) What are the outcomes and what is the significance of outcomes
Our reply:
Thanks for all the above comments, which are valuable to improving our work's organization and readability. We added a paragraph at the end of Section 1 to summarize our work’s contributions. We have reorganized our manuscript a lot to address its survey nature in ANS-related techniques and applications. Please refer to our reply to comment 4 for the details.
- From lines 70 to 777, (Section 2, Section 3, Section 4.1, Section 4.2 Section 4.3) contain a lot of background information on ANS. While the understanding of basic working of ANS is important to understand this article, it is really useless to consumes almost 700 (which is almost three fourth (3/4 or 75 percent) of the entire article length. Therefore, it is highly recommended to remove all the unnecessary details. The basic working of ANS and its applications can be summarized in a single section.
Our reply:
- In facing the fact that only a few members in the compression academia community and the domestic industry have noticed the progress of ANS-related techniques, we think a thorough overview of ANS is beneficial. At the same time, we agree with the reviewer's comments about the poor organization of the original manuscript in descriptions of ANS-related backgrounds.
- To enhance the readability of our work, we moved the illustrative simple for explaining the process of non-uniform ABS encoding to Appendix A.
- We moved the pseudo-codes and illustrative examples for both ANS Stream Encoding and Decoding to Appendix B to understand better the renormalization process mentioned initially in Section 3.2(b).
- d. We moved the concrete and step-by-step example for illustrating the complete tANS processes (as summarized initially at the end of Section 3.3(c)) to Appendix C.
.
- A literature review section (Section 3) is needed to understand the current compression methods in digital images. How ANS is different from the current compression methods.
Our reply:
We agree with the reviewer’s comment that providing a detailed review of current compression methods in digital images would benefit the readers. However, it seems hard for us to complete a thorough survey in such a short period. Instead, we conducted a series of experiments in compressing well-known benchmark images by using well-adopted lossless compression algorithms to show their performance differences in compression ratio and execution speed (Please refer to Section 6 of the revision for details). This way may justify that ANS behaves differently from other well-known entropy coding algorithms.
- Section 4 should describe the Proposed ANS-based Digital Image Processing System. Figure 4.1, 4.2 and 4.3 should be elaborated in more detail. Section 5 should describe the experimental results. This section is really the place where authors can describe that how they have implemented the solution (ANS to digital images). It may contain the following subsection:
- Experimental setup
- Description of tools and particular settings used in the experiment
- Description of Benchmarks and case study.
- Experimentation Procedure
- Experimental Results: summarize the results in some tabular or graphical form and discuss in details.
- Section 6 should provide the Performance Comparison and Discussion. Compare your results (obtained in the previous section) with state-of-art solutions, using tables, graphs and any other methods for your comparison. Good data visualization techniques may enhance the value of your article. Discuss each figure, table or chart in the text (detailed description). Briefly that what is the purpose and message of each figure or table. Use various performance parameters for comparisons. Each performance parameter must be defined explicitly before its use. You should also provide the justification and motivation behind the selection of each performance parameter. It is VERY IMPORTANT to highlight the strengths of your article as compared to existing methods. Discuss the reasons that why you were able to obtain good results. Similarly, it is equally important to mention the shortcomings of your solution with respect to current methods. Discuss the reasons for these limitations and shortcomings.
Our reply:
As suggested, we add a new Section (Section 6) in the revision to address our experimental settings and report our experimental results for comparing nearly all well-known lossless compression algorithms with benchmark testing images usually used in the image communities. All experimental results are clearly shown graphically and summarized in tables in detail. Moreover. as suggested, performance comparisons and discussions comparing our results with state-of-art solutions are also included. Finally, a paragraph is added at the end of Section 7 to address the possible research directions of our future work, which somewhat indicates the limitations and shortcomings of the current ANS-related methods can achieve.
Minor Issues:
- Abstract should be in the form of a single paragraph. Done!
- The formatting is not consistent. Done, as much as possible!
- Article should be checked for typos. Done, as much as possible!
Round 2
Reviewer 1 Report
I'm satisfied with the answers.
Reviewer 2 Report
Authors have addressed the comments.
There are no further concerns from my side.